# A Semi-Automatic Tool for the Standardized Analysis of Fluorescent Intensity Changes in Polarized Cells

**DOI:** 10.3390/ijms26209987

**Published:** 2025-10-14

**Authors:** Fruzsina Fazekas, Tibor Zelles, Eszter Berekméri

**Affiliations:** 1Department of Oral Biology, Semmelweis University, 1089 Budapest, Hungary; 2Department of Zoology, University of Veterinary Medicine Budapest, 1078 Budapest, Hungary; 3Department of Pharmacology and Pharmacotherapy, Semmelweis University, 1089 Budapest, Hungary; 4Laboratory of Molecular Pharmacology, HUN-REN Institute of Experimental Medicine, 1083 Budapest, Hungary

**Keywords:** single cell, Ca^2+^ imaging, ROI determination

## Abstract

Imaging of intracellular messengers, like calcium, is one of the most reliable methods to follow real-time changes in several aspects of cellular activity, like receptor activation. However, the analysis could be influenced and biased by several factors like the location, shape, and size of the regions of interest (ROIs) and by the detection and correction of the movement of the preparation. Programs which are provided by the manufacturers are expensive and cannot be shared by collaborators. Many self-made programs have been implemented lately which have in-built cell recognizer ROI identification functions. These programs focus on the soma of the cells and neglect the processes, because in full tissue preparation finding cells is still challenging. Subcellular imaging experiments are still rare. To the best of our knowledge there is no program which can automatically define ROIs for subcellular imaging experiments even in single indicated cells with complex morphology. We developed and validated a program to address this gap using simple and understandable mathematical methods for ROI determination and simple statistics for movement correction. Validation experiments were conducted on cochlear Deiters’ cells. Deiters’ cells have processed morphology which connects two fluid compartments in the cochlea. Because of the function and the fine morphology of the cell, it could be interesting to examine the subcellular Ca^2+^ handling mechanisms of it. Test impulses were activated by ATP. With some limitations the program successfully fulfilled its purpose. As a free, easily understandable, and open-source program, we hope it will help to analyze and plan subcellular experiments.

## 1. Introduction

To follow intracellular mechanisms in real time, one of the most powerful tools is the monitoring of the cells’ ion concentration with fluorescent dyes or proteins (e.g., [1,2,3]). Calcium (Ca^2+^) indicators are the most well spread of ion indicators in research. They reliably measure ion concentration and, thanks to the constant development, are available in several forms, from chemical indicators to genetically modified molecules [4,5,6,7]. The Ca^2+^ affinity of the indicators varies widely, which makes it possible to choose the best one to answer the research questions. Changes in the emitted fluorescent intensity, indicating the Ca^2+^ chelation of the indicators, could be followed and recorded by a (high-speed) camera. A multitude of programs have been developed to analyze recordings of intensity changes; however, reproducibility remains poor [8,9,10].

Firstly, much of the software provided by companies loses its support over the years and cannot be used in longer periods (which happened in our laboratory). Reanalysis, or checking previous results are not possible because of this. Programs with longer support usually cost more, and raw data cannot be shared with cooperators unless they buy it too [11,12]. Code sources are usually not open, although it could help to adjust the analysis to a special cell type. Because of these reasons, several home-made analyzer tools were developed by different laboratories, with an increasing number of codes becoming available on the internet [13,14,15,16,17,18,19,20,21,22,23,24,25] (for a short comparation see Appendix A). There are programs which can handle the extracted signals too [26].

To reproduce an analysis, the subjective components should be minimalized. In case of fluorescent intensity measurements, the parameters of the area to be analyzed, i.e., the region of interest (ROI), can highly influence the results. If ROIs are determined on the recordings manually, there is only a minimum probability that two different people, or even the same person at two different times, would measure the exact same intensities (even if the results are very similar and differences are not relevant in each case). In most of the newly implemented programs, the determination of the ROIs is performed by the algorithm, which eliminates the subjectivity [13,14,15,16,17,18,19,20,21,22,23,24,25]. However, these programs are made for field recordings, where many cells (even hundreds) are investigated and only somatic activity is measured. Separately indicated cells with special morphology are still measured by manually located ROIs [27,28,29]. Moreover, subcellular imaging studies are also influenced by the shape, size, and location of the ROIs. One of the most interesting phenomena in many cells is the spread of the intracellular Ca^2+^ concentration changes, which form Ca^2+^ waves in the cells [30,31,32,33]. For investigating these waves, more ROIs should be placed on the image of the cell, with the differences in intensity measured within these regions then being compared. Up to now, we have had no knowledge about a program which can locate objectively more regions on a separated cell to measure the subcellular Ca^2+^ activity of it.

We aimed to prepare software that can objectively and reproducibly measure the intensity changes in subcellular activity of polarized, not neural cells, for example, retinal cones or rods and cochlear Deiters’ cells (DCs). The processes of the cells are sometimes reported to have a different Ca^2+^ handling mechanism compared to the cell body. Our program successfully tested on DCs. It inserted ROIs in the same size and shape and included a movement detection to eliminate movement artifacts.

## 2. Results

### 2.1. Region of Interest Identification for Subcellular Ca^2+^ Imaging in Deiters’ Cells

Cochlear Deiters’ cells (DCs) are highly polarized cells with one apical process (Figure 1A) that undergoes changes in morphology during postnatal development. Cochlea morphology slightly varies in mice strains so 2 different mouse strains (BALB/c and CD1) across a wide age spectrum (from 5 to 22 days old) were tested to make sure the program can handle the different cell anatomy [34,35]. The anatomy of the cochlear cells also differs according to their location (due to the frequency specificity). In order to validate the program regarding these anatomical variations, recordings of cells from the apical, middle, and basal turns of the cochlea were used. It is evident that these three turns are responsible for the sensation of different sound frequencies (low, mid-range, and high, respectively). Furthermore, these are the three locations where cells can be observed in a hemicochlear preparation which is literally a cochlea cut in half along the midmodiolar plane.

Firstly, we tested how the human factor affects the ROI definition (Figure 1). For this, two experienced analyzers (who worked with Deiters’ cells for at least 3 years) analyzed 15 Deiters’ cells loaded with Oregon Green BAPTA (OGB) dyes by single-cell electroporation. The following 3 measurements were compared: (1) with ROIs that had been delineated along the outline of the cell body and the uppermost part of the process (hereinafter referred to as “structure following ROIs”), used typically for whole cell activation analysis; (2) 5 equally sized ROIs, one of them includes the top of the process and 4 located on the soma of the cell; (3) 9 equally sized ROIs, one of them includes the top of the process and 8 located on the soma. The last two are used for analyzing intracellular Ca^2+^ waves and subcellular activation of the cell. The elevated ROI number improves precision but elevates the complication of the analysis.

The differences found in the ROIs (Figure 1B), the measured amplitude (Figure 1C), and the width of the response (Figure 1D) of the two analyzers were compared with linear models. There was not a single measurement which was the same in case of the two analyses; however, the results of the experienced analyzers did not differ significantly.

The program made by our research group identifies ROIs similar to the last measured scenario of the manual experiments: nine equally sized ROIs, one located on the top of the process and same-sized ROIs on the soma. This is the most human-sourceful and time-needing scenario where the most errors could occur but allows for the most accurate intracellular analysis. The difference is that the program defines one ROI on the top of the process, and the same-sized and -shaped ROIs are defined on the soma; however, instead a fixed number of ROIs (e.g., nine as in the manual experiments), it defines as many as it fits on the surface of the soma (Figure 2).

Our experiments were performed in hemicochlea preparation. In this organotypic preparation other structures (membranes, other cells) are also observed next to the chosen cells. During an imperfect single-cell electroporation indicator loading, the environment can be stained. In some cases, this background noise could be eliminated by our cleaning technique (Figure 3). However, we must mention that in instances when the amount of dye in the environment was higher, the cleaning was not successful in removing the loading bias, resulting in unreliable ROI identification (Figure 3). In isolated cells these problems are not relevant.

Our program uses the orientation of the cell to help in the process–soma difference detection: it projects the number of values equal to 1 in the given columns to the x-axis of the plot, formed from the intensity matrix. This means that the original orientation of the cell influences the success of the ROI definition. In one case the program failed to find the process, when the cell lined vertically (Figure 4A,B); however, a little tilt and the cleaning is enough to find the difference between the process and the soma (Figure 4C,D).

### 2.2. Motion Detection and Correction

Our experiments used hemicochlea preparation [36]. Its three-dimensional morphology makes it difficult to localize it in the chamber, and sometimes the perfusion interacts with the preparation, moving it away (Figure 5A,E,F). Due to this movement, the cell may drift out of the pre-selected area to be examined on the recording; therefore, it could cause significant changes in the detected fluorescent intensity. To prevent the artificial decrease in the amplitudes, manual readjustment of the analyzed regions is needed, which makes the analyzing process even more labor-intensive. Our program includes a built-in movement sensing and ROI adjusting subprogram (Figure 5). It compares the intensity of the pixels between the ROI measuring the background intensity and the ROIs located on the cell. If the difference is not significant, it relocates the ROIs. The significance level can be changed by the user depending on the needed precision and the available time. During the relocalization of the ROIs the size and shape remain the same as previously defined.

We compared the program-generated amplitudes and response durations with the mean of the analyst-recorded values (Figure 6). After testing the normality of the data distribution, *t*-tests or nonparametric Wilcoxon tests were used to assess the differences. No significant differences were detected in any case.

### 2.3. Other Cell Types

Our laboratory’s (at the Oral Biology Department of Semmelweis University) main profile is the Ca^2+^ imaging of the cochlear Deiters’ cell. However, we wanted to make a program which is suitable for other cell types, with similar, processed morphology. For this we sourced databases with Ca^2+^ imaging experiments of other cells. Salamander cone cells were found, and their outer light sensitive part can be handled as a process of the cell [27]. Appendix A pone.0006723.s002.avi file was converted to .tiff file and the scales were removed with ImageJ (version 1.54). Our program defined the ROIs well on the salamander cone cells (Figure 7A—1st layer of the tiff file with the defined ROIs) and measured the intensity changes as well (Figure 7B). Because of the different ratios of the cells (bigger process) there were less ROIs located on the soma.

Since we did not find other full experiment movies about cells, we could only test our program’s ROI determination feature on fluorescent images of cells with processes. Individually labeled astrocytes [37,38,39], odontoblasts [40], and myoblast cells [41] were tested successfully (Appendix A). Our program can find the cells if they are separable from the environment, meaning it could be used not only for Ca^2+^ indicators, but any other indicator type is suitable for the detection.

Depending on the question, astrocytes with richer arborization could also be investigated with our program, even if the location of the ROIs does not perfectly align with the cell’s axis (Appendix A).

## 3. Discussion

Ca^2+^ imaging is one of the oldest (since 1970s) and most reliable methods to measure and follow cell reactions to different outer and inner stimuli [42,43,44]. Next to the constantly developed dyes and genetically encoded molecules, the analysis could be influenced by the ROI definition (Figure 1). Programs manufactured by companies usually work with closed code systems, which are expensive and difficult to share with collaborators. New, self- (and laboratory-) made programs are formed to help the cooperative work, usually with open-source code that could be modified to focus more on the task, even with minimal coding abilities [13,17,18]. The automatic ROI definition is a basic criteria for the reproducibility of the experiment and the analysis, because even experienced analysts localize ROIs differently (Figure 1). However, as most of the Ca^2+^ research is performed in neurobiology and focuses on more cells in the same section of the field, the main ROI definition programs are made to identify cells [13,14,15,16,17,18,19,20,21,22,23,24,25]. Subcellular imaging experiments are rare, with the biggest developments in this field focused on neural process structure [45,46,47] and astroglia process territories [48,49]. These programs are investigating full tissue preparations without the visibility of the whole cell, and cellular parts cannot be measured. As far as we know, no program was made for identifying multiple ROIs on the same cell which is fully visible. Cells with processed morphology often exhibit different Ca^2+^ handling mechanisms in this compartment, e.g., elongated or higher concentration elevation (Figure 7) [27]. These subcellular differences could be interesting and worth investigating in more detail.

We successfully developed and validated the first open-sourced program for the subcellular analysis of changes in the intensity of whole cells with process. Our test cells were cochlear Deiters’ cells, located under the outer hair cells (receptor cells) and reaching with their process towards the apical site of the receptor cells [50]. The soma and the process are located in two different compartments of the cochlea: the process reaches the endolymph whilst the soma baths in the perilymph, two different solutions with different signal sources. Trivially, different reactions are expected from the compartments, but no objective analysis was presented to analyze these. For instance, endolymph’s ATP concentration elevates faster after a noise trauma, so ATP should activate endolymphatic surface structures more intensively [51]. Our program uses the process’s intensity and location to form the basic ROI shape, and the same-sized and -shaped ROIs are localized on the soma from the apical part to the basis of it (Figure 2). The movement of the preparation is also corrected, and the tolerance level can be changed by the user depending on the expected motion of the preparation and the time source (Figure 5). Motion correction is also a built-in plug-in in most of the programs that help in the analysis of Ca^2+^ imaging recordings, based on fast Fourier transformation [52,53], hidden Markov models [54] or other techniques (based on the Lucas–Kanade framework: [55,56], template-matching: [20,57], more different algorithms: [58] or self-made [22]). Compared to these motion detections, our approach is much simpler: it compares pixel intensities inside the ROI on the cell and in the background. During the validation (Figure 5) this easier and faster mathematical process performed well.

Our program was tested on other processed cells: salamander cone cells [27] which performed similarly well to the imaging of the Deiters’ cells (Figure 7). We have to mention that we had tried to find more cells and contacted authors who had similar publications to test our program on a wider range of cells; however, the search for cells was unsuccessful and no responses were received from the authors. We would also like to encourage researchers to share their data to help the development of science. Instead of videos, simple anatomical pictures were used to test the ROI determination of the cells. With one processed cell (like odontoblast [40] or elongated myocytes [41]) which oriented longitudinally (without U-turns in the soma) our ROI determination works well (Appendix A).

The program still has some limitations that are influenced by the experimental protocols. At this moment, the program cannot handle multiple visible cells in the recordings, only a single indicated cell can be measured within the field of view. In the case of isolated cells, the program works perfectly. In organ preparations, like hemicochlea used by our laboratory, the indicator load could be imperfect, resulting in the additional loading of nearby anatomical structures with dye or the remaining of the indicator in the background (Figure 3). These additional light pixels are counted in the midline calculation and therefore influence the locations of the ROIs. However, if only a small amount of dye entered the environment, the image cleaning steps of the program could handle this bias (Figure 3). The orientation of the cell in the field of view can also influence the success of the ROI determination (Figure 4). A slight tilt is required to separate the soma from the cell’s process correctly.

Cells with bigger arborization (like astrocytes) were also tested, but the program is not designed for this type of process. However, we hypothesize that a transcellular Ca^2+^ wave could be investigated by the ROIs resulted by our program (Appendix A), if the arborization is localized evenly around the cell body. However, we do not think this would be the optimal program for astrocyte examination.

We are planning to continuously improve the program, to be more perfect to analyze cells with more processes and branching arborization so that it would be a good choice to examine both single astrocytes and neurons. With the help of our program, we hope single cell and subcellular Ca^2+^ imaging studies will be more widespread in the future.

Our program’s code is freely available on GitHub at https://github.com/berekmerieszter/SiCeAn/tree/main (accessed on 8 October 2025).

## 4. Materials and Methods

### 4.1. Collection of Experimental Data from Ca^2+^ Imaging of Deiters’ Cells

All the experimental procedures and the animal care were in accordance with the National Institute of Health Guide for the Care and Use of Laboratory Animals. The procedures were approved by the Animal Use Committee of Semmelweis University, Budapest (ethical approval number PE/AE/1912-7/2017).

BALB/c and CD1 mice strains of both sexes were sacrificed between post-natal day 5 (P5) to P22, in all three turns of the hemicochlea preparatum (apical, middle, and basal turns). The BALB/c mouse is among the most widely used genetically homogeneous inbred strain used in biomedical research, while CD1 mice are a commonly used outbred mouse stock with a high degree of genetic variability. Performing experiments on widely used mice strains with different genetic diversity strengthens the validation of the new method. The experiment protocol was based on previous studies [59]. Briefly, the mice were decapitated under isoflurane anesthesia, cochleae were removed and placed in an ice-cold oxygenated perilymph-like solution (composition in mM: NaCl 22.5, KCl 3.5, CaCl_2_ 1, MgCl_2_ 1, Hepes 10, Na-gluconate 120, glucose 5.55 (all from Sigma-Aldrich, Budapest, Hungary); pH 7.4; 320 mOsm/L). In the cutting chamber of a vibratome (Vibratome Series 3000, Technical Products International Inc., St. Louis, MO, USA) cochleae were halved along the midmodiolar plane. Hemicochleae were placed into an imaging chamber filled and perfused (speed 3.5 mL/min) with the oxygenated experimental solution. The cells of interest were selected under a LUMPlanFl 40×/0.80 w water immersion objective (Olympus, Tokyo, Japan) with oblique red illumination and were loaded by single-cell electroporation. For electroporation, the borosilicate pipettes (5–7 MΩ; Harvard Apparatus, Holliston, MA, USA) were filled with light excitable Ca^2+^ indicator Oregon Green 488 BAPTA-1 or -6F hexapotassium salt (OGB-1 or OGB-6F) (ThermoFisher Scientific, Waltham, MA, USA) dissolved in distilled water (final concentration of 1 mM). The pipettes were mounted onto an electrode holder, which was attached to a micromanipulator (Burleigh PCS-5000, Thorlabs, Munich, Germany). When pipette approached the cell of interest, a single square wave current impulse (10 ms, 10 μA) was applied, allowing the cell to be loaded with the indicator. The current was generated by a pCLAMP10 software-guided stimulator system (Biostim STE-7c, Supertech Ltd., Pecs, Hungary; MultiClamp 700B Amplifier and Digidata 1322A, Molecular Devices, Budapest, Hungary).

The calcium imaging procedure was performed at room temperature, similar as we described earlier. Excitation light of 494 ± 5 nm (Polychrome II monochromator, TILL Photonics, Planegg, Germany) was used to illuminate the cells, and the light emitted was recorded after passing through a band-pass filter (523 ± 25 nm). An Olympus BX50WI fluorescence microscope (Olympus, Tokyo, Japan) equipped with a Photometrics Quantix cooled CCD camera (Photometrics, Tucson, AZ, USA) was used and was controlled with the Imaging Workbench 6.0 software (INDEC BioSystems, Los Altos, CA, USA). The images were taken at a frame rate of 1 Hz during the drug-evoked responses, and the rest of the time, the frame rate was 0.1 Hz to reduce phototoxicity and photobleaching. In the perfusion chamber, the volume of the buffer was about 1.9 mL. The cells were stimulated by 100 µM ATP (Sigma-Aldrich, St. Louis, MO, USA), and added to the perfusion for 30 s. Before ATP stimulation, a baseline period of at least 3 min was registered.

Fluorescence intensities were calculated in R 4.5.1. version program. Background-corrected using a nearby area was devoid of loaded cells. The relative fluorescence changes were calculated as follows: dF/F_0_ = (F_t_ − F_0_)/F_0_, where F_0_ is the fluorescence intensity of the baseline and Ft is the fluorescence intensity at time t. The maximal amplitude and the duration of the transients (calculated as the time between 50% of the maximal amplitude at the increasing and descending part of the response). For the statistical analysis generalized linear regression model (from lme4 package, version: 1.1-37) was used with experiment ID as a random factor.

### 4.2. Structure and Operation of the Algorithm

The algorithm was written in the R programming language (version: 4.5.1.), the R packages used in this project include grid (version: 4.4.0), imager (version: 1.0.5), quantmod (version: 0.4.26), resahpe2 (version: 1.4.4), scales (version: 1.3.0), and tiff (version: 0.1-12).

The input file must be in TIFF multipage format (for some more details about the .tiff file see Appendix A). Before the initiation, a series of functions need to be executed, which together form the primary algorithm: kneedle [60,61], image cleaner, rotation, ROI maker, ROI calculator, and inflect. The detailed operation of these functions is described in the subsequent sections. A schematic overview of the algorithm’s operation is presented in Figure 8. At the end of the program, the results are stored in the form of a data frame to facilitate further analysis and data handling. The exported data include the frame index (which image in the sequence is being processed), the timepoint at which the image was captured (if the tiff metadata includes it), the mean intensity measured within the ROI placed on the process, the mean intensities of the ROIs on the soma, the mean intensity of the background ROI, and a logical indicator of whether the ROIs were recalculated for that frame.

#### 4.2.1. Start

After launching the program, a logical check is performed to determine whether the algorithm is processing the first frame. In order to optimize runtime, most steps are executed only during the analysis of the first frame. The values calculated at this stage are largely applicable to subsequent frames, thereby avoiding redundant computations for each new frame. If the current frame is the first one, the program proceeds to the function named image cleaner.

#### 4.2.2. Kneedle

Kneedle function [60,61] is part of the image cleaner labeled as “Detecting elbow point” on the flowchart (Figure 8). This is one of the two mechanisms in the program that was not independently developed; instead, an existing algorithm, which was made available by Emmet Tam, was incorporated (the other mechanism is the inflect function).

This function is necessary to identify the intensity value that separates the intensities of the cell and that of the background. The function takes a curve as an input and finds the so-called “elbow” point (Figure 9) of it (see Appendix A for further detail). The coordinates of the elbow point are then returned as the output of the function and can be further utilized in subsequent steps.

#### 4.2.3. Image Cleaner

Image cleaner transforms the pixel intensity values into the image matrix so that only binary values (0 and 1) remain. In an ideal scenario, this results in the cell body being entirely assigned the value of 1, while the background receives only 0s.

A histogram (usually a unimodal curve) is first generated from the pixel intensity values of the input image matrix (Figure 10). Cropped regions of the recording (with intensity values of 0, applied for storage efficiency) (Figure 4A and Figure 11) were excluded from the histogram data. The maximum represents the background intensity levels, as these (typically black–gray values) constitute the majority of the data in case of single cell investigations. The cell body is represented by the flattening section of the curve beyond the peak and the second elbow point; therefore, only the portion of the curve following the mode is passed to the kneedle function (Figure 10). This is a mandatory step, given that the kneedle function is capable of detecting only one elbow point.

The kneedle function returns the coordinates of the elbow of the modified histogram, which determines the intensity threshold: all intensity values below this are set to 0, while those above are converted to 1. The resulting image matrix is then visualized by the image cleaner function, which prompts the user to indicate whether further adjustment is necessary. If the answer is negative, the image matrix is saved, and the program progresses to the next step. If the response is affirmative, additional values are set to zero if they lack at least three neighboring cells with a value of one (possibly noise pixels in the background, based on Conway’s game of life [62]). Thereafter, the program displays the updated matrix and again prompts the user about further adjustment.

#### 4.2.4. Calculation of the Rotation Angle

To eliminate overlap between the columns containing the process intensities and those containing the cell body values, we rotate the cell into a roughly horizontal position. The program fits a regression line to the image matrix values obtained after the 0–1 separation (further information can be found in the Appendix A). The slope of the fitted line is then used to derive the required angle for rotation, as the regression line has proven to be sufficiently accurate for this purpose.

#### 4.2.5. Rotation

The rotation of the image is performed with the help of the imager R package (https://cran.r-project.org/web/packages/imager/imager.pdf (accessed on 10 September 2025)). The rotation is achieved by applying the imrotate() function (the image matrix is converted to a cimg object for this step), utilizing linear interpolation.

#### 4.2.6. Examination of Process Orientation

Following the rotation by the angle corresponding to the slope, the Deiters’ cell can be oriented in two distinct ways, depending on its original position. The process can thus be located either to the right or to the left of the cell body. This condition is verified by the inflect function which finds the local maxima of the curves formed again from the coordinates of the pixels with values of 1. For the purposes of the program, the process must be positioned to the right of the cell body. If the smaller local maximum (representing the process) is on the left of the bigger local maximum (that represents the soma), the program rotates the cell by 180° and adapts the rotation angle to meet the requirements. At the end of this procedure, the correct rotation angle is saved separately so that it can be directly applied to subsequent layers without recalculating.

#### 4.2.7. ROI Maker

The ROI maker function is responsible for determining the positions of the ROIs. This function, like the others, is not invoked for every frame after the first, but only if the program detects a displacement.

The function first identifies the cell and places an initial ROI over the process, sized to cover most of its extent. Subsequently, additional ROIs are arranged along the midline of the cell body, all with identical dimensions to ensure comparability (for a more detailed description of these steps see Section S4 of the Detailed Methods in the Appendix A). The quantity of ROIs therefore depends on the number of process-sized regions that can be accommodated along the cell (Figure 11). In addition, for reference purposes an ROI is placed in the background. At the end of the procedure, the function saves the ROI positions on the original image.

#### 4.2.8. ROI Calculator

This function appears in the flowchart as the shape labeled “Mean intensity of ROIs”. Its sole purpose is to save the intensity values within the ROIs that have been calculated and positioned by the ROI maker, along with their mean intensity. This allows them to be used in subsequent analyses.

#### 4.2.9. Motion Detection

This part of the program is designed to ensure that the ROIs remain correctly positioned. During experiments, the cell often drifts from its original position, potentially moving out of the regions covered by the ROIs. Therefore, it is important to correct any motion when necessary. To keep the runtime as optional as possible, the check is only performed on every fifth layer, where the program compares the median intensities within the ROIs of the cells with those of the background using Wilcoxon signed rank test (see Appendix A for more detail).

#### 4.2.10. Saving

Finally, the program stores the layer number, the corresponding time, the average intensities within the ROIs on the process, the cell body, and the background. In addition, it also stores an indicator of whether ROI positions were recalculated for that layer due to drift. This information is stored in the form of a data frame, so that it can easily be saved to a file.

## 5. Conclusions

Ca^2+^ imaging is one of the most widely used methods in cell physiology. Several programs have been developed in individual laboratories to identify cells in tissue preparations, correct motion artifacts, and extract intensity curves. However, no software has yet been developed for the detailed analysis of individual cells and their subcellular compartments.

Our research group has developed an open-source program for the analysis of individually processed, morphologically identified cells. The software can objectively define ROIs, correct tissue displacement, and track and save intensity curves. The program, written in R, is intended to support the analysis of intracellular Ca^2+^ waves, for which no sufficiently objective tool has been available so far.

## Figures and Tables

**Figure 1 ijms-26-09987-f001:**
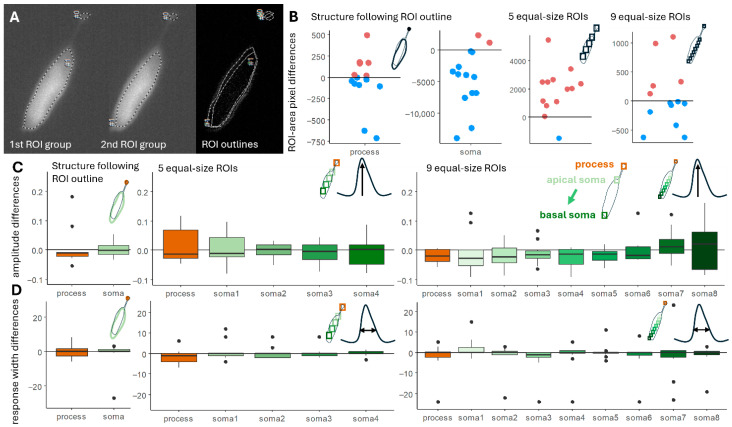
Manually located ROIs caused differences. ROIs were defined in three different ways as follows: structure following ROIs include the top of the process and one ROI on the cell’s soma; 5 and 9 equal sized ROIs which are used for intracellular measurements, with one ROI including the process and the remaining ROIs located on the soma in balanced distances. (**A**) Example for structure following ROIs in case of the two analyzers, (**B**) Differences in the area of the two analyzers’ ROIs in pixels. The horizontal zero line shows the ideal case in which the two ROIs are the same in terms of the number of pixels, whilst blue points indicate when the second analyzer drew bigger ROIs, and pinkish points indicate the instances in which the second analyzer drew smaller ROIs compared to the ROIs of the first analyzer. In case of structure following ROIs, ROIs on the processes are more similar compared to the ones on the somas (see the differences in the y axis); small ROIs are more similar in case of the equal ROIs, too. (**C**) Measured normalized amplitude (dF/F0) differences during the two analyses: ideal 0 difference indicated by the line, process values are brownish, soma values are greenish. No significant differences were detected between the two analyzers’ measurements; however, variance is bigger at the end of the soma where the location has more chance to include more background pixels. (**D**) Response width (the time from the elevating 50% of the maximal intensity to the decreasing 50% of it) varied less; however, many outliers were detected. The ideal 0 difference between the analyzers’ measurements is indicated by the line. Process response width is indicated by brownish color; soma values are greenish.

**Figure 2 ijms-26-09987-f002:**
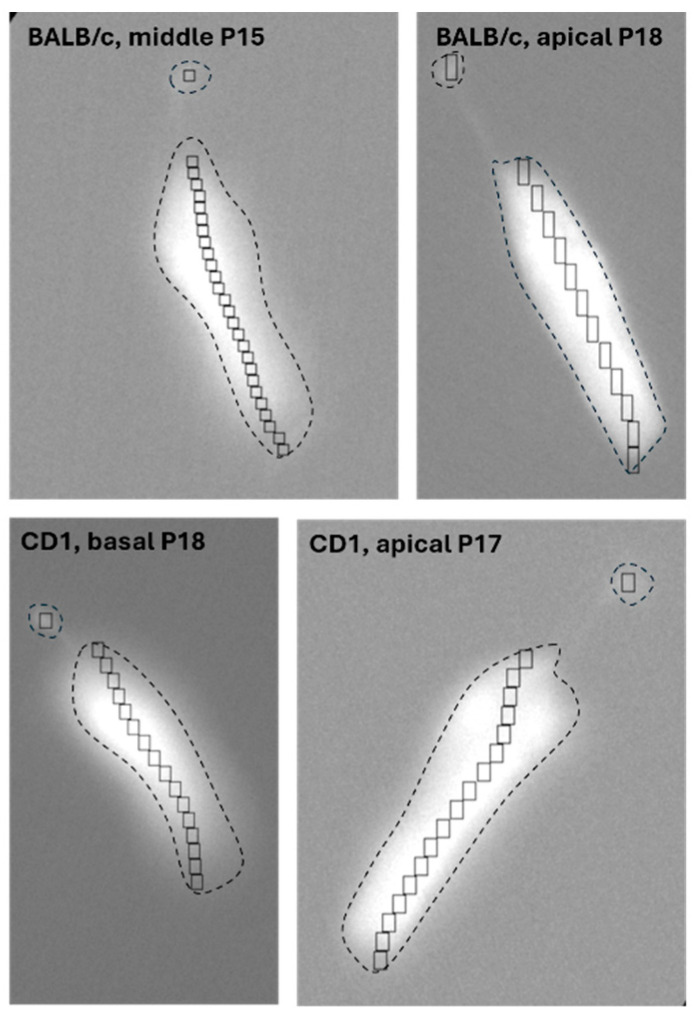
Successful ROI identification in different Deiters’ cells. For intracellular measurements, the objective ROI definition was accurate with our program. One ROI was inserted on the top of the process of the cell, whilst several ROIs of the same size and shape were inserted on the soma, as many as could fit along the cell’s axis. We used different strains (CD1 and BALB/c) and age (from 5 to 22 postnatal days old) of mice and measured multiple turns which all present different morphology according to the frequency map.

**Figure 3 ijms-26-09987-f003:**
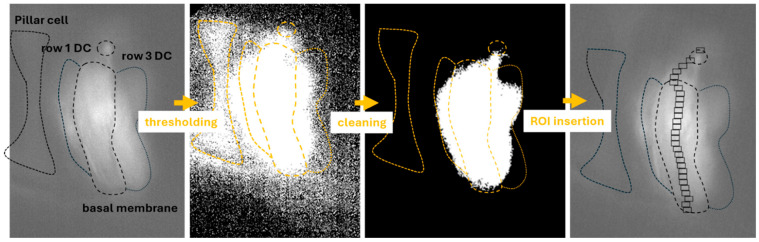
ROI identification of imperfectly loaded midrow Deiters’ cells. Deiters’ cells form 3 lines in the organ of Corti, with close contact with each other. During the single-cell electroporation dye-load, neighboring cells could also be dyed; additionally, the basal membrane and the pillar cells nearby are also prone to bind dye molecules. The program first thresholds the images (see Section 4.2.3 of the Materials and Methods) deciding whether the intensity of each pixel should be 0 or 1. It may even elevate the noise of the image as seen in this example. However, the next cleaning steps eliminate the intense pixels without neighbors, leaving only the most intensive structure visible (here it still includes more than 1 cell). The ROI identification steps locate the ROIs in the midline of the brightest structure.

**Figure 4 ijms-26-09987-f004:**
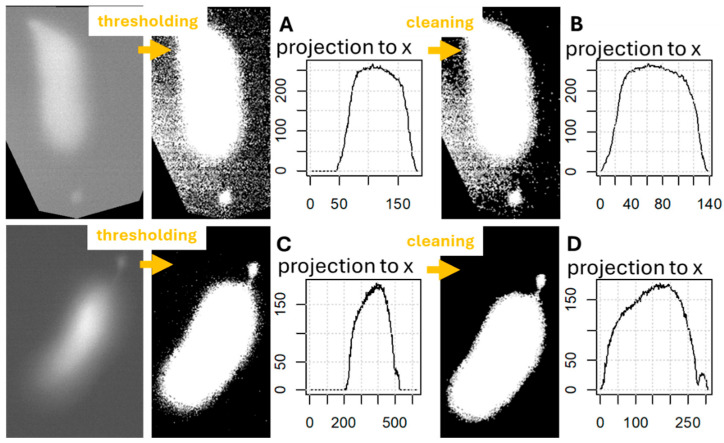
Process identification depends on the cell orientation in the field of view. Our program failed to find the process orientation only in one case, when the cell was oriented vertically in the field. The workflow (**A**,**C**) firstly sets a threshold, such that all pixel values above it are set to 1 and beneath it to 0, then calculates the sum of the pixels in each column of the image-represented matrix. If 2 local maxima could be detected, the process is distinguishable from the soma. Pixel cleaning process (**B**,**D**) can—but not always—help in the identification. The vertically located cell’s process was not found after the cleaning (**B**); however, with a small tilt, the lower cell’s process was identified after cleaning (**D**), but not without it (**C**).

**Figure 5 ijms-26-09987-f005:**
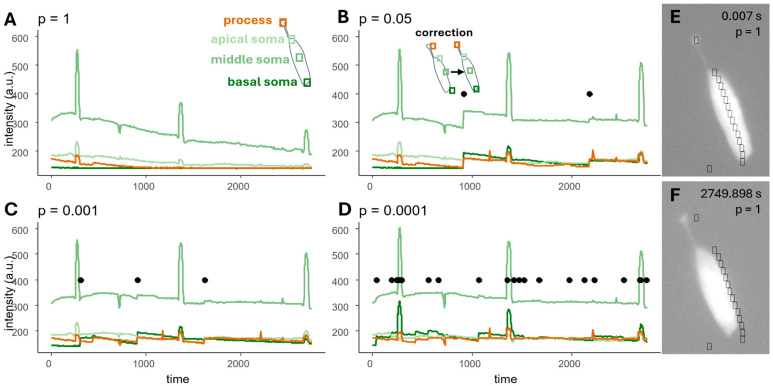
Movement sensing and ROI location adjustment are built in the program. During a longer experiment, movement of the preparation could happen; example (**E**) shows the original position of the cell where the program defines the ROIs and (**F**) shows the location of the same cell and the previously defined ROIs more than 2700 s later. (**A**) The intensity curves measured without adjusting the location of the ROIs decrease constantly during the experiment, as the cell moves away. Only four ROIs’ intensity curves are shown—the differently colored curves were measured at different locations on the cell’s body, as indicated in the small cell figure (orange: head of the process, light green: first ROI on the soma, green: ROI on the middle of the soma, dark green: last ROI on the soma). The ATP-induced amplitudes are also influenced by the movement of the baseline. The program can relocate the ROIs according to a set up threshold ((**B**)—0.05, (**C**)—0.001, and (**D**)—0.001). The timepoints at which adjustments were made are registered and presented by black dots on the figures. The difference between the adjusted and unadjusted ROI measurements are visible, and have huge influence on the registered amplitudes, too.

**Figure 6 ijms-26-09987-f006:**
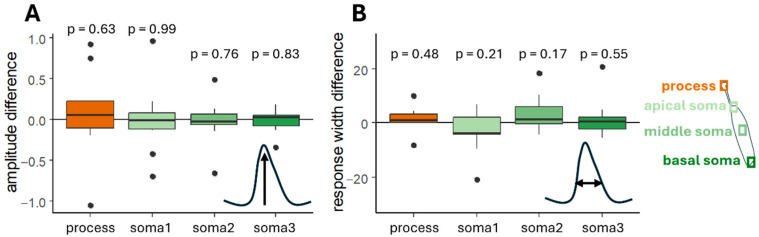
Program-generated curves compared with the mean of the manually analyzed curves. Neither the amplitudes (**A**) nor the durations of the responses (**B**) showed significant differences. The generated curves were background-corrected, and dF/F_0_ values were calculated. Maximal values were recorded as amplitudes, and the time between 50% of the amplitude on the rising and falling phases was defined as the width of the response. Differences between the program and manual recordings were calculated and tested for normality. In cases of normal distribution, a *t*-test was used; otherwise, the Wilcoxon test was applied. Calculated *p*-values are indicated on the panels. No significant differences were detected.

**Figure 7 ijms-26-09987-f007:**
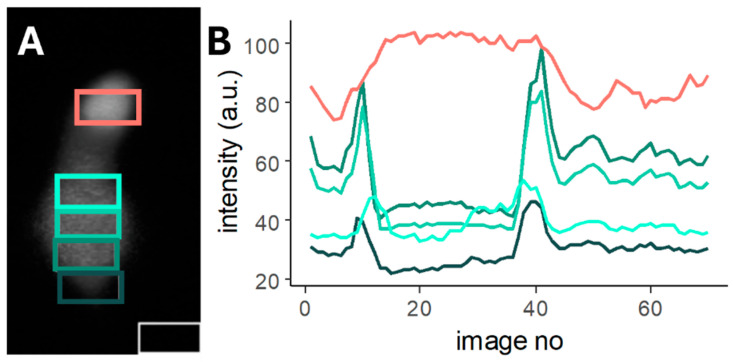
Salamander cone cells’ ROIs are well defined by the program. (**A**) A cone cell found in a Appendix A, with the program-defined ROIs on the cell, each ROI colored differently. The original recording was modified so as to cut down the intensity scale with ImageJ. Because of the different ratio of the process/soma, the number and size of the ROIs were visibly different compared to a Deiters’ cell ROI group but met the criteria (localization and same size). (**B**) The intensity curves that were measured inside the ROIs located on the cone cell (background intensity not shown). The coloration of the curves is consistent with the coloration of the ROIs: the reddish curve was measured within the ROI on the process, while the differently shaded aquamarine curves were measured within the ROIs on the cell body.

**Figure 8 ijms-26-09987-f008:**
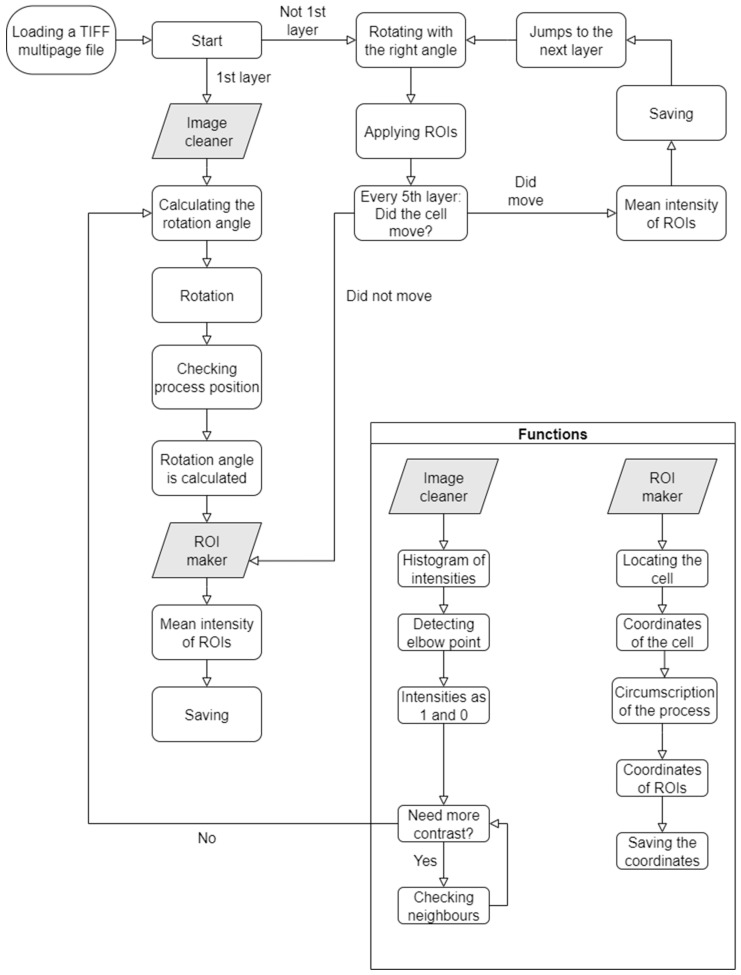
Flowchart summarizing the operation of the program. The two most complex functions, Image cleaner and ROI maker, are highlighted separately with their respective steps indicated.

**Figure 9 ijms-26-09987-f009:**
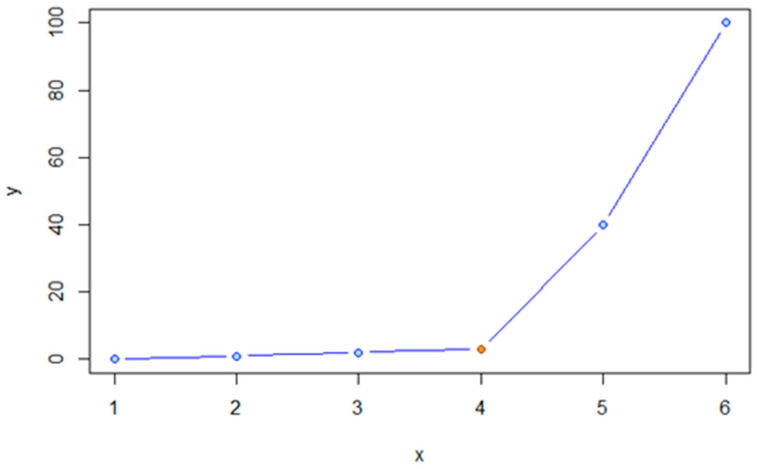
Illustration of the elbow. In this figure, the elbow is indicated by the red dot. The original image was sourced from the algorithm’s developer, with the red marking added subsequently for visual emphasis.

**Figure 10 ijms-26-09987-f010:**
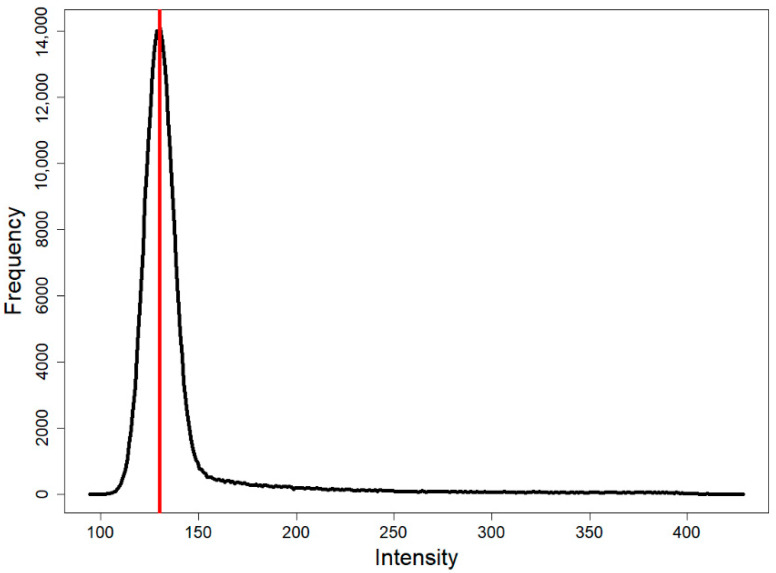
Histogram generated from the image pixel intensity values. The red line indicates the mode value, excluding pixels with an intensity of 0. The kneedle algorithm receives only the data points to the right of the red line, ensuring that the resulting curve contains a single elbow point.

**Figure 11 ijms-26-09987-f011:**
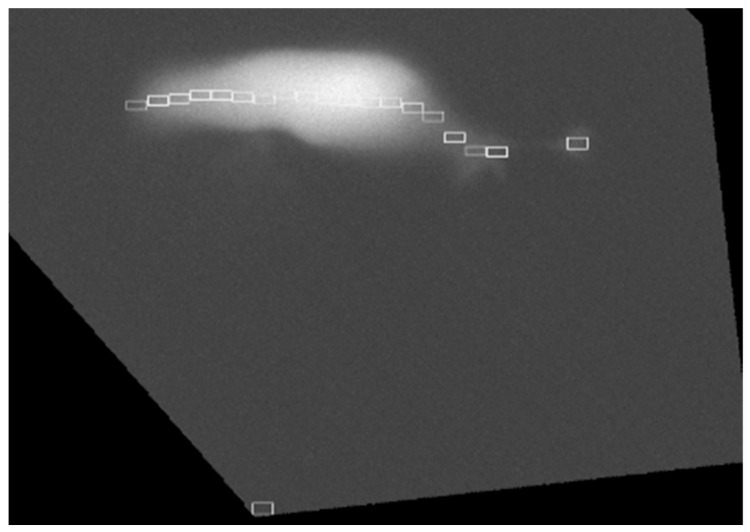
ROI maker forms ROIs based on the process details. The first ROI is located on the process. ROIs (represented by the white rectangles) are determined from the right to the left, so the process is the first where the program starts to form the ROI. The program can properly locate the ROIs only on the head of the process and the soma (as usual there is an intensity gap between the process and the soma). The ROI on the background (for reference purposes) is located far from the cell to avoid any influence by the cell’s emitted light.

## Data Availability

All used experimental data are available by asking (berekmeri.eszter@univet.hu).

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
