# Peer review of "A Semi-Automatic Tool for the Standardized Analysis of Fluorescent Intensity Changes in Polarized Cells"

_ijms, 2025, doi:10.3390/ijms26209987_

Round 1

Reviewer 1 Report

Comments and Suggestions for Authors

Observations (academic style, bullet points)

  • The manuscript addresses an important methodological gap in calcium imaging, particularly in the objective identification of subcellular ROIs in polarized cells.

  • The topic is timely and relevant, given the limitations of commercial and existing open-source software for reproducible ROI analysis.

  • The novelty lies in combining ROI definition with motion correction in a freely available and customizable tool.

  • The introduction provides a broad background but occasionally reiterates information, which may dilute the central research question.

  • The description of algorithmic steps is highly detailed, demonstrating transparency, but risks overwhelming the reader when included in the main text.

  • The results convincingly show that manual ROI selection introduces variability even among experienced analysts.

  • The automated approach presented improves consistency and allows for more precise subcellular calcium wave analysis.

  • The validation is carried out on Deiters’ cells and salamander cones, showing adaptability across morphologically distinct cell types.

  • The scope of validation remains limited, as only a small number of cell types and experimental conditions are tested.

  • The figures are informative, but in some cases too dense or complex for immediate comprehension without extended reference to the text.

  • Legends could provide more detail to ensure independent interpretability of the figures.

  • The discussion section emphasizes the strengths of the program but could better contextualize the contribution relative to other available software.

  • Limitations are mentioned (e.g., inability to analyze multiple cells simultaneously, dependence on cell orientation, background noise issues), but they are not deeply analyzed.

  • The manuscript successfully demonstrates proof of concept but does not fully quantify the performance gains compared with existing methods.

  • The availability of the code via GitHub is a strong aspect, fostering reproducibility and community adoption.

  • The conclusion highlights the utility of the tool but could be more nuanced in outlining its limitations and possible future improvements.

Recommendations (academic style)

  • Condense the Introduction to focus more sharply on the research gap and novelty of the current work.

  • Relocate highly technical algorithmic descriptions to Supplementary Materials, while retaining conceptual clarity in the main text.

  • Expand the validation by including additional cell types and experimental conditions to support broader applicability.

  • Incorporate benchmarking metrics (e.g., reproducibility indices, accuracy rates, error quantification) to objectively demonstrate the tool’s performance.

  • Provide a structured comparison with existing open-source software, clarifying the distinct contributions of this program.

  • Simplify figures for clarity and ensure legends are sufficiently comprehensive for independent interpretation.

  • Enhance the discussion by critically reflecting on how the improved ROI definition impacts biological interpretations, particularly calcium wave dynamics.

  • Elaborate on limitations and provide concrete suggestions for future developments, such as multi-cell analysis, orientation-independent ROI detection, or integration with large-scale imaging datasets.

  • Refine the conclusion to balance achievements with limitations, and to clearly indicate directions for future research and software evolution.

  • Revise the manuscript for linguistic precision and fluency, ensuring a consistent academic style throughout.

Author Response

Responses to reviewer 1

We thank the reviewer for the precise review and for the valuable comments and recommendations that helped us to improve our manuscript. Please find our detailed responses below:

  • The introduction provides a broad background but occasionally reiterates information, which may dilute the central research question.
  • Condense the Introduction to focus more sharply on the research gap and novelty of the current work.

Thank you. We reread and revised the Introduction to make it more focused and understandable, reducing redundancy and emphasizing the research gap and novelty.

  • The description of algorithmic steps is highly detailed, demonstrating transparency, but risks overwhelming the reader when included in the main text.
  • Relocate highly technical algorithmic descriptions to Supplementary Materials, while retaining conceptual clarity in the main text.

Our intention was to ensure clarity and transparency so that researchers can easily use our program. Following your advice, we revised the algorithm description section, retaining only the key steps in the main text and moving the detailed descriptions to the Supplementary Materials.

  • The scope of validation remains limited, as only a small number of cell types and experimental conditions are tested.
  • Expand the validation by including additional cell types and experimental conditions to support broader applicability.
  • Incorporate benchmarking metrics (e.g., reproducibility indices, accuracy rates, error quantification) to objectively demonstrate the tool’s performance.

Thank you for this important suggestion. We also intended to test more cell types; however, we could not find suitable imaging videos of single cells. While many recordings exist for brain-slice studies, such image series fall outside the scope of this manuscript. To avoid incomplete validation, we tested the program’s ROI determination features using anatomical images of odontoblasts, myoblasts, and astrocytes. Two new supplementary figures (Supplementary Figures 1 and 2) illustrate these results.

ROIs of cells with elongated structures or a single process were successfully determined (Supplementary Figure 1). For highly branched cells or those with U-shaped processes, ROI localization was less precise but still useful depending on the research question (e.g., studying intracellular Ca²⁺ waves when processes are evenly distributed around the astrocyte soma).

We also added a new Figure 6 comparing program-generated and analyst-generated curves. The amplitudes and durations of Ca²⁺ responses did not differ significantly across subcellular regions analyzed.

  • The figures are informative, but in some cases too dense or complex for immediate comprehension without extended reference to the text.
  • Legends could provide more detail to ensure independent interpretability of the figures.
  • Simplify figures for clarity and ensure legends are sufficiently comprehensive for independent interpretation.

Thank you for this suggestion. We improved Figures 1, 2, 5, and 6 by adding small icons and rearranged the panels in Figure 7. We believe these changes improve clarity and make the figures more easily interpretable.

  • The discussion section emphasizes the strengths of the program but could better contextualize the contribution relative to other available software.
  • Provide a structured comparison with existing open-source software, clarifying the distinct contributions of this program.

Thank you for this valuable recommendation. Many excellent open-source programs exist for tracking Ca²⁺ indicator intensity changes and extracting curves from recordings, a topic worthy of a dedicated review. Within our manuscript’s constraints, we summarized their common features (e.g., ROI determination, motion correction) and highlighted their limitations regarding subcellular intensity analysis. We also added Supplementary Table 1, summarizing key aspects of these programs.

  • Limitations are mentioned (e.g., inability to analyze multiple cells simultaneously, dependence on cell orientation, background noise issues), but they are not deeply analyzed.
  • Elaborate on limitations and provide concrete suggestions for future developments, such as multi-cell analysis, orientation-independent ROI detection, or integration with large-scale imaging datasets.
  • The conclusion highlights the utility of the tool but could be more nuanced in outlining its limitations and possible future improvements.
  • Refine the conclusion to balance achievements with limitations, and to clearly indicate directions for future research and software evolution.

Thank you for the insightful suggestion. In future work, we plan to focus on orientation-independent ROI determination and improve the program’s ability to recognize more processes and branches. These developments will enable more detailed investigations of astrocytes. We added a new Supplementary Figure 2 illustrating potential improvements for non-optimally determined ROIs.

  • The manuscript successfully demonstrates proof of concept but does not fully quantify the performance gains compared with existing methods.
  • Enhance the discussion by critically reflecting on how the improved ROI definition impacts biological interpretations, particularly calcium wave dynamics.

Thank you for the comment. To our knowledge, no existing software can determine subcellularly localized ROIs and analyze individual cellular compartments. We emphasized and elaborated on this feature in the Discussion section.

  • Revise the manuscript for linguistic precision and fluency, ensuring a consistent academic style throughout.

Thank you, we reread and tried to eliminate our errors.

We hope that the revised manuscript meets the reviewer’s expectations. We are grateful for the constructive comments and suggestions, which significantly improved the quality of our work.

Reviewer 2 Report

Comments and Suggestions for Authors

The manuscript presents a new analysis software for calcium imaging. A few suggestions to the authors:

-Mention the ethical approval number and date in the M&M section.

-Explain the rationale for using BALB/c and CD1 mice.

-What do the authors think about whether the method and software described could be extended to other calcium probes with different spectra or maybe other intracellular targets (e.g., ROS) using appropriate probes?

-Developing an analysis software of this complexity requires expertise in computer science, specifically software engineering, and none of the authors' affiliations reflect that. Please elaborate on that aspect.

-What plans do the authors have for this software? Is it going to be available to researchers worldwide free of charge? One of the limitations the authors highlighted in the currently available programs is that they are expensive and cannot be shared among collaborators.

-The Abstract is too general and 90% of it is only stating the problem over and over again while what was actually achieved, how it was achieved, and what implications it might have, is only succinctly mentioned. 

-How does the newly developed software compare to existing alternatives? What advantages does it provide? This must be clearly mentioned in the Abstract and in the Discussion/Conclusion.

-The manuscript contains many grammatical errors and awkward phrasing and must therefore be revised for clarity and accuracy by a professional service.

Author Response

Responses to reviewer 2.

We thank the reviewer for the valuable comments and suggestions that have helped us to improve our manuscript. Our detailed responses are provided below:

-Mention the ethical approval number and date in the M&M section.

Thank you. We have added the ethical approval number and date to the Materials and Methods section.

-Explain the rationale for using BALB/c and CD1 mice.

Thank you for this suggestion. Several studies have investigated morphological differences in cochlear cells between various mouse strains. We have included this rationale and supporting references in the revised text.

-What do the authors think about whether the method and software described could be extended to other calcium probes with different spectra or maybe other intracellular targets (e.g., ROS) using appropriate probes?

Thank you for this interesting question. We believe that the program can indeed be used with different indicators. For additional validation, we analyzed anatomical images (newly added Supplementary Figures 1 and 2). In these cases, intensity dynamics were not measured (as only single images were used), but ROI determination worked effectively. The key condition for successful ROI detection is a sufficient intensity contrast between the cell and its surroundings, which can be achieved with any suitable indicator.

-Developing an analysis software of this complexity requires expertise in computer science, specifically software engineering, and none of the authors’ affiliations reflect that. Please elaborate on that aspect.

Thank you for the observation. We agree that developing a well-designed, user-friendly program typically requires extensive software engineering experience. Two of the authors have programming skills acquired through self-learning and formal courses, and also possess a strong self-taught mathematical background. Many existing open-source programs use highly complex algorithms that are difficult to adapt without advanced training in mathematics or computer science. In contrast, our algorithm relies on simpler mathematical and statistical methods rather than techniques such as Hidden Markov Models or Fast Fourier Transformations. Although this may limit some aspects of the program, it makes the code easier to understand, modify, and apply—even for researchers without extensive computational backgrounds.

-What plans do the authors have for this software? Is it going to be available to researchers worldwide free of charge? One of the limitations the authors highlighted in the currently available programs is that they are expensive and cannot be shared among collaborators.

Thank you for this question. Our program is already publicly available on GitHub, free of charge. We will continue to update it by uploading new versions, user manuals, and documentation. We are also committed to providing ongoing user support and incorporating feedback from the research community.

-The Abstract is too general and 90% of it is only stating the problem over and over again while what was actually achieved, how it was achieved, and what implications it might have, is only succinctly mentioned. 

Thank you for pointing this out. We have rewritten the Abstract to be more informative and concise, highlighting the main objectives, methods, results, and implications of the study.

-How does the newly developed software compare to existing alternatives? What advantages does it provide? This must be clearly mentioned in the Abstract and in the Discussion/Conclusion.

Thank you for the suggestion. We expanded the Discussion and Conclusion sections to emphasize the advantages of our software, and we added a new Supplementary Table 1 comparing our program with existing alternatives. The main novelty of our software lies in its ability to determine subcellular ROIs, which is essential for compartmental Ca²⁺ imaging studies. To our knowledge, no currently available program offers this capability.

-The manuscript contains many grammatical errors and awkward phrasing and must therefore be revised for clarity and accuracy by a professional service.

Thank you for the suggestion. We carefully reread the entire manuscript, corrected grammatical errors, and improved overall clarity and readability.

We hope that these revisions address all of the reviewer’s concerns. We are grateful for the constructive feedback, which has significantly enhanced the quality and clarity of our manuscript.

Reviewer 3 Report

Comments and Suggestions for Authors

1) Line 12 “different stimuli.” Mention some potential stimuli

2) Explain if the investigation could be used for different chemical compounds or only for Ca2+

3) Lines 45-48, why this part of the manuscript has 15 references “[13, 14, 23–28, 15–22]” Includes only essential references. Not more than 5

4) Line 57 has many references “[13, 14, 23–28, 15–22]” try to reduce these references

5) Some parts of the introduction have many references, and some parts of this section don’t have references. Improve the introduction part

6) Line 79 “mouse strains (BALB/c and CD1) across a wide age spectrum” try to include age information (days or weeks)

7) Lines 151-152 “Our program uses the location of the cell to help in the process-soma difference detection” explain the advantages and disadvantages of using your method

8) Line 168, include a reference for this sentence “Our experiments used hemicochlea preparation which is literally a cochlea cut in half.

9) Figure 5 was difficult to understand, because it has some results with different colors, but does not have information about the colors green, orange, black circles, etc. Try to improve it

10) Line 193 “Our laboratory main profile is the Ca2+ imaging of the cochlear Deiters’ cell.” Include the name information about this laboratory

11) Lines 195-196, include information about the importance of studying “databases with Ca2+ imaging experiments

12) Figure 6, the same recommendation as Figure 5

13) Line 212 “Ca2+ imaging is one of the oldest”, include information on the year to initiate the investigation by this method

14) The discussion part has again the same group of references (lines 219 and 223) “[13, 14, 23–28, 15–22]”. Try to use adequate references

15) Line 229 includes scheme for this mechanism “different Ca2+ handling mechanisms in the process”

16) Lines 236-237 “Trivially, different reactions are expected from the compartments” this sentence is very general, try to improve it, include information about these “different reactions”

17) Line 296 why “The calcium imaging procedure was performed at room temperature”

18) The manuscript has some interesting results, but doesn’t have a discussion. Improve the discussion for all

19) Optional, try to improve the figure quality

20) Includes conclusions

Author Response

Responses to reviewer 3.

We thank the reviewer for the valuable comments and suggestions, which have helped us to further improve the quality of our manuscript. Our detailed responses are provided below:

1) Line 12 “different stimuli.” Mention some potential stimuli

Thank you. We have added examples of potential stimuli to this sentence.

2) Explain if the investigation could be used for different chemical compounds or only for Ca2+

Thank you for this comment. We have also included Supplementary Figures 1 and 2, where anatomical indicators (staining) were used on different cell types. The condition for the program’s ROI determination feature is the intensity difference between the cell and its environment, which can be achieved with any suitable indicator or fluorescent antibody.

3) Lines 45-48, why this part of the manuscript has 15 references “[13, 14, 23–28, 15–22]” Includes only essential references. Not more than 5

4) Line 57 has many references “[13, 14, 23–28, 15–22]” try to reduce these references

14) The discussion part has again the same group of references (lines 219 and 223) “[13, 14, 23–28, 15–22]”. Try to use adequate references

Thank you for the suggestion. This group of references represents the collection of laboratory-developed programs for Ca²⁺ imaging analysis. We prefer not to remove any of these works, as our aim is to acknowledge all researchers who contributed to this field. Currently, no comprehensive review summarizes these tools. To address this point, we have added a brief comparative summary in Supplementary Table 1.

5) Some parts of the introduction have many references, and some parts of this section don’t have references. Improve the introduction part

Thank you. We have revised the Introduction to ensure a more balanced use of references and made it more focused and precise.

6) Line 79 “mouse strains (BALB/c and CD1) across a wide age spectrum” try to include age information (days or weeks)

Thank you. We have added this information: mice ranged in age from 5 to 22 days old.

7) Lines 151-152 “Our program uses the location of the cell to help in the process-soma difference detection” explain the advantages and disadvantages of using your method

Thank you. We added a detailed explanation in the Detailed Methods section. The main disadvantage is that ROI determination depends on cell orientation, which we plan to address in future developments. The main advantage is that this method is straightforward and visually intuitive, even for users without strong mathematical or engineering backgrounds. Many open-source programs remain underutilized due to algorithmic complexity. We believe true open-source accessibility includes comprehensibility, allowing researchers to adapt the software to their needs.

8) Line 168, include a reference for this sentence “Our experiments used hemicochlea preparation which is literally a cochlea cut in half.

Thank you. We have inserted a reference for this statement.

9) Figure 5 was difficult to understand, because it has some results with different colors, but does not have information about the colors green, orange, black circles, etc. Try to improve it

12) Figure 6, the same recommendation as Figure 5

19) Optional, try to improve the figure quality

Thank you. We have improved Figures 2, 5, 6, and 7 by enhancing color clarity, adding legends and small icons, and optimizing layout for better readability.

10) Line 193 “Our laboratory main profile is the Ca2+ imaging of the cochlear Deiters’ cell.” Include the name information about this laboratory

Thank you. We have added the name of the laboratory to this section.

11) Lines 195-196, include information about the importance of studying “databases with Ca2+ imaging experiments

Thank you. We elaborated on this section, explaining that we reviewed databases and supplementary materials to find suitable Ca²⁺ imaging recordings for validating our program.

13) Line 212 “Ca2+ imaging is one of the oldest”, include information on the year to initiate the investigation by this method

Thank you. We have included the appropriate reference and year in the revised text.

15) Line 229 includes scheme for this mechanism “different Ca2+ handling mechanisms in the process”

Thank you. We have provided an illustrative example of this mechanism.

16) Lines 236-237 “Trivially, different reactions are expected from the compartments” this sentence is very general, try to improve it, include information about these “different reactions”

Thank you. We have rewritten this sentence to be more specific and have added an illustrative example to clarify the differences among compartmental responses.

17) Line 296 why “The calcium imaging procedure was performed at room temperature”

Thank you for the question. Our laboratory protocol is performed at room temperature following preparation (which uses ice-cold solutions). The main reason is practical rather than biological: cells remain viable for approximately two hours at room temperature, which is sufficient for our experiments.

18) The manuscript has some interesting results, but doesn’t have a discussion. Improve the discussion for all

Thank you. We expanded the Discussion section to be more comprehensive, focused, and analytical.

20) Includes conclusions

Thank you. We have added a concise Conclusions section summarizing the main findings and implications of our study.

We hope these revisions satisfactorily address the reviewer’s comments. We appreciate the detailed feedback and constructive suggestions, which have substantially improved our manuscript.

Round 2

Reviewer 2 Report

Comments and Suggestions for Authors

The authors have adequately addressed my comments. The language remains an issue.

Comments on the Quality of English Language

Must be improved.